**Data Availability Statement:** All relevant data are within the manuscript and its Supporting Information files.

# Association between *Trypanosoma cruzi* DTU TcII and chronic Chagas disease clinical presentation and outcome in an urban cohort in Brazil

**Marco Antonio Prates Nielebock**[1☯], **Otacílio C. Moreira**[2☯], **Samanta Cristina das Chagas Xavier**[3], **Luciana de Freitas Campos Miranda**[4], **Ana Carolina Bastos de Lima**[2], **Thayanne Oliveira de Jesus Sales Pereira**[1], **Alejandro Marcel Hasslocher-Moreno**[1], **Constança Britto**[2], **Luiz Henrique Conde Sangenis**[1], **Roberto Magalhães Saraiva**[1]*

1 Clinical Research Laboratory in Chagas Disease, Evandro Chagas National Institute of Infectious Diseases, Fiocruz, Rio de Janeiro, Brazil, 2 Molecular Biology and Endemic Diseases Laboratory, Oswaldo Cruz Institute, Fiocruz, Rio de Janeiro, Brazil, 3 Trypanosomatid Biology Laboratory, Oswaldo Cruz Institute, Fiocruz, Rio de Janeiro, Brazil, 4 Leishmaniasis Surveillance Laboratory, Evandro Chagas National Institute of Infectious Diseases, Fiocruz, Rio de Janeiro, Brazil

☯ These authors contributed equally to this work.
* roberto.saraiva@ini.fiocruz.br

## Abstract

### Background

The specific roles of parasite characteristics and immunological factors of the host in Chagas disease progression and prognosis are still under debate. *Trypanosoma cruzi* genotype may be an important determinant of the clinical chronic Chagas disease form and prognosis. This study aimed to identify the potential association between *T. cruzi* genotypes and the clinical presentations of chronic Chagas disease.

### Methodology/principal findings

This is a retrospective study using *T. cruzi* isolated from blood culture samples of 43 patients with chronic Chagas disease. From 43 patients, 42 were born in Brazil, mainly in Southeast and Northeast Brazilian regions, and one patient was born in Bolivia. Their mean age at the time of blood collection was 52.4±13.2 years. The clinical presentation was as follows 51.1% cardiac form, 25.6% indeterminate form, and 23.3% cardiodigestive form. Discrete typing unit (DTU) was determined by multilocus conventional PCR. TcII (n = 40) and TcVI (n = 2) were the DTUs identified. DTU was unidentifiable in one patient. The average follow-up time after blood culture was 5.7±4.4 years. A total of 14 patients (32.5%) died and one patient underwent heart transplantation. The cause of death was sudden cardiac arrest in six patients, heart failure in five patients, not related to Chagas disease in one patient, and ignored in two patients. A total of 8 patients (18.6%) progressed, all of them within the cardiac or cardiodigestive forms.

**Funding:** This work was supported by Coordenação de Aperfeiçoamento de Pessoal de Nível Superior (CAPES/PROEX, finance code 001; PrInt Fiocruz-CAPES), Education Ministry, Brazil and by Fundação de Amparo à Pesquisa do Estado do Rio de Janeiro (FAPERJ), Brazil [grant number 210.497/2019 to Dr. RM Saraiva] - Financial Disclosure: Dr. C Britto and Dr. OC Moreira – researcher fellows of CNPq and FAPERJ (CNE, JCNE).

**Competing interests:** The authors declare that they have no competing interests.

## Conclusions/significance

TcII was the main *T. cruzi* DTU identified in chronic Chagas disease Brazilian patients (92.9%) with either cardiac, indeterminate or cardiodigestive forms, born at Southeast and Northeast regions. Other DTU found in much less frequency was TcVI (4.8%). TcII was also associated to patients that evolved with heart failure or sudden cardiac arrest, the two most common and ominous consequences of the cardiac form of Chagas disease.

## Introduction

Chagas disease (CD) is one of the tropical neglected diseases recognized by the World Health Organization (WHO) and has a great public health and socioeconomic burden in endemic countries [1]. Around 6 to 7 million people are chronically infected by the protozoan *Trypanosoma cruzi* worldwide [2], from whom 5.7 million live in Latin America, mainly in Argentina, Bolivia, Brazil, Colombia, and Mexico [2, 3]. Sixty to seventy percent of the chronically infected patients do not present any clinical evidence of organ damage due to Chagas disease and present the clinical indeterminate form of the disease, but 30 to 40% present the cardiac, digestive or cardiodigestive forms of the disease [4]. The reasons pointed out for this pleiotropic presentation are various and include factors from both the host and the parasite.

Nowadays, the high genetic variability of the *T. cruzi* allows its classification into seven different lineages, as follows: six DTUs (Discrete Typing Units), from TcI to TcVI, and a seventh genotype, TcBat [5–7]. Initially identified in several bat species, TcBat was found in a child in Colombia [8]. Any *T. cruzi* lineage can infect humans, however TcI, TcII, TcV, and TcVI are the DTUs mostly associated to human infections in domicile cycles transmission in endemic areas [9, 10]. In Brazil, TcII and TcVI are the *T. cruzi* DTUs most frequently identified in human infections [11–14].

Regarding the association between *T. cruzi* genotypes and clinical presentations of chronic Chagas disease, TcI, TcII, TcIV, TcV, and TcVI were identified in patients with chronic cardiac form born in Argentina, Brazil, Bolivia, Colombia, and Venezuela [11, 15–23], while TcII, TcV, and TcVI were also identified in patients with the digestive form, particularly in Brazil, Argentina, and Bolivia [16, 22–24]. TcIV seems to have a secondary importance in patients with Chagas cardiomyopathy in Colombia and Venezuela [19–21]. On the other hand, some studies were unable to show conclusive evidence of the association between a specific DTU and Chagas heart disease [11, 17], and TcIII is usually found in sylvatic cycles and was identified in patients with the chronic indeterminate form in Brazil [25].

In this paper, we describe the *T. cruzi* DTU genotypes of blood culture isolates obtained from 43 patients followed at our outpatient clinic in order to correlate the DTU with the clinical presentation and the place of birth.

## Methods

### Patients and study design

This is a retrospective study that used a convenience sample formed by all positive *T. cruzi* blood culture from adult patients from both sexes regularly followed at the outpatient center of the Evandro Chagas National Institute of Infectious Diseases (INI) between July 2008 and June 2010. All patients had Chagas disease previously diagnosed by two simultaneously positive serological tests (indirect immunofluorescence and ELISA) (S1 Table). All participants who

were still followed at our institution were approached during their regular medical appointments and provided written informed consent allowing the use of their blood culture samples and granting access to their medical records. The institutional ethics committee waived the requirement for informed consent for deceased participants and those who were lost to follow-up and could not be reached. Clinical, epidemiological and mortality data were obtained from medical records. Mortality data were also retrieved from registries of death certificates available at the department of justice of the Rio de Janeiro state (http://www4.tjrj.jus.br/SEIDEWEB/default.aspx). Final follow-up date was arbitrarily defined as of June 2019. Chagas disease clinical form was classified according to the II Brazilian Consensus on Chagas disease from 2015 [3] using electrocardiographic, 2D Doppler echocardiographic, upper and lower gastrointestinal endoscopic, and contrast radiographic exams available in medical records.

## Ethical approval

This study was approved by the Evandro Chagas National Institute of Infectious Diseases Ethical Committee under number 62973116.6.0000.5262. All procedures followed regulatory guidelines and standards for research involving human beings as stated in the Brazilian National Health Council Resolution 466/2012 and were conducted according to the principles expressed in the Declaration of Helsinki in order to safeguard the rights and welfare of the participants.

## Blood culture

Blood culture was performed in biphasic culture medium Novy-MacNeal-Nicolle medium plus Schneider's Drosophila Medium (Sigma-Aldrich, St. Louis, Missouri, USA) supplemented with 10% inactivated fetal bovine serum and antibiotics 200 IU penicillin and 200 μg/mL streptomycin, as previously described [26]. The culture tubes were incubated at 26–28˚C in a biochemical oxygen demand (BOD) incubator and examined every 15 days for up to 60 days. The parasites isolated in culture were cryopreserved in liquid nitrogen ($N_2L$). Growth of parasites was performed in sterile bottles for cell culture in the same biphasic culture medium. The total culture volume obtained was centrifuged at 7000 rpm for 10 minutes and the pellet was submitted to three washes in NaCl-EDTA buffer to obtain the parasite mass, which was stored in a freezer at –20˚C until DNA extraction to carry out molecular techniques.

## DNA extraction

DNA extraction from the pellet of parasite isolates was done using silica columns using the High Pure PCR Template Preparation (Roche, Germany) kit following previously published protocol [27]. At the last stage of the protocol, DNA was eluted in 100 μL of elution buffer and stored at -20˚C until use.

## Molecular typing

*T. cruzi* genotyping into DTUs from I to VI was performed as reported [28], following a combination of methodologies previously described based on multilocus conventional PCR [11, 19, 29]. As a panel of positive controls, we used *T. cruzi* epimastigotes from subpopulations classified as DTUs TcI to TcVI (clones/strains: Dm28c (TcI), Y (TcII), INPA 3663 (TcIII), INPA 4167 (TcIV), LL014 (TcV), and CL (TcVI)), obtained from the Protozoan Collection of the Oswaldo Cruz Foundation (Colprot), were used as reference. The PCRs targeted the intergenic region of Spliced Leader (SL-IRac) [UTCC and TCac primers], to distinguish between TcI (150 bp), TcII, V or VI (157 bp) and TcIII or TcIV (200 bp), (SL-IR I and II) [TCC, TC1 and TC2 primers] [30, 31], to distinguish between TcI (350 bp), TcII, TcV and TcVI (300 bp) and TcIII and TcIV (not

**Table 1. Primers for *Trypanosoma cruzi* molecular typing.**

| Target | Primers | Sequence [5'– 3'] |
|---|---|---|
| SL–IRac | UTCC | CGTACCAATATAGTACAGAAACTG |
| | TCac | CTCCCCAGTGTGGCCTGGG |
| SL–IR I and II | TCC | CCCCCCTCCCAGGCCACACTG |
| | TC1 | GTGTCCGCCACCTCCTTCGGGCC |
| | TC2 | CCTGCAGGCACACGTGTGTGTG |
| 24Sα–rDNA | D75 | GCAGATCTTGGTTGGCGTAG |
| First round | D76 | GGTTCTCTGTTGCCCCTTTT |
| 24Sα–rDNA | D71 | AAGGTGCGTCGACAGTGTGG |
| Second round | D76 | GGTTCTCTGTTGCCCCTTTT |
| A10 | Pr1 | CCGCTAAGCAGTTCTGTCCATA |
| First round | P6 | GTGATCGCAGGAAACGTGA |
| A10 | Pr1 | CCGCTAAGCAGTTCTGTCCATA |
| Second round | Pr3M | CGTGGCATGGGGTAATAAAGCA |

amplified)], the D7 domain of the 24Sα ribosomal RNA gene [Heminested PCR: D75 and D76 (first round) and D76 and D71 (second round), to distinguish between TcII and TcVI (140 bp), TcIII (125 bp), TcIV (140/145 bp) and TcV (125 or 125+140 bp)], and the A10 nuclear fragment [Heminested PCR: Pr1 and P6 (first round) and Pr1 and Pr3 (second round), to differentiate TcII (690/580 bp) from TcVI (630/525 bp)] [32, 33] (Table 1 and Fig 1).

The amplification reactions were performed in a Veriti Thermal Cycler (Applied Biosystems), as follows: 5 μL of extracted DNA were added to a 12,5 μL GoTaq Green Master Mix 2X (Promega, Madison, USA) containing GoTaq DNA polymerase, buffer (pH 8.5), 400 μM of each dNTP and 3 mM $MgCl_2$, 1.25 μL of each primer (stock solutions: 25 μM for the SL-IR target, 10 μM for the 24Sα and A10 targets), and 5 μL of ultrapure water. PCR products (25 μL) were separated by agarose gel electrophoresis (3.0% w/v, 90V, 1 hour), stained with Gel Red (Biotum) 0.1 X and visualized at UV light.

## Map construction

Georeferencing of each patient was performed from the centroid of the municipality, using the online cartographic platform Google Earth, with the geodetic reference system WGS 84

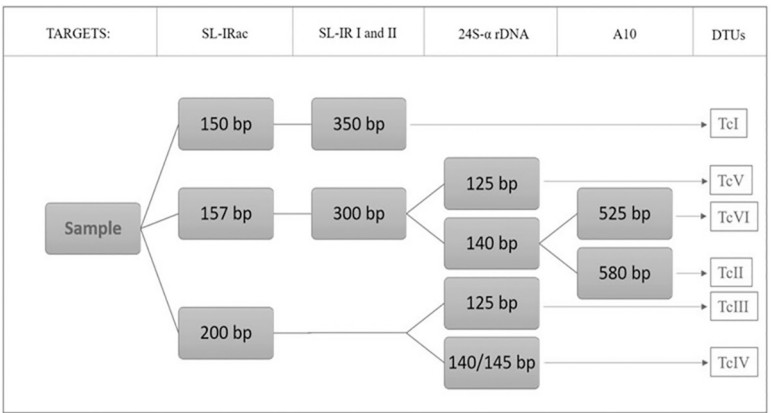

**Fig 1. Characterization targets flowchart for *T. cruzi* multilocus conventional PCR and expected sizes of amplified products, based in four molecular markers [28].**

(World Geodetic System 1984) (S1 Table). For the map construction of the distribution of the *T. cruzi* genotypes, points of the samples localization were visualized in a Geographic Information System (GIS) in the Quantum GIS software version 3.4 (Madeira), using the continental, national, and State boundaries, extracted from the open access (public domain) cartographic base of Brazilian Institute of Geography and Statistics (IBGE) accessed at https://www.ibge.gov.br/geociencias/downloads-geociencias.html.

### Statistical analyses

All statistical analyses were performed using MedCalc 12.5.0.0. software. Continuous variables were expressed as mean ± standard deviation (sd) and categorical variables as absolute and percentage values.

## Results

### Patients characteristics

A total of 43 patients presented blood culture positive for *T. cruzi* between July 2008 and June 2010. Most of these patients were born in rural areas of the Southeast and Northeast Brazilian regions. The number of cases from Southeastern states were as follows, Minas Gerais (n = 6), Rio de Janeiro (n = 1), and São Paulo (n = 1) and the number of cases from Northeastern states were as follows, Bahia (n = 14), Pernambuco (n = 11), Paraíba (n = 5), Sergipe (n = 2), and Alagoas (n = 1). Only one patient was born in state of Mato Grosso do Sul, in Midwest Brazilian region. No patient was born in the North or South Brazilian regions. One patient was not natural from Brazil but was from the city of Santa Cruz de La Sierra, located at the Prurinacional state of Bolivia. Most patients were women (72.1%) and were infected by vector borne transmission (90.7%) (Table 2).

Most patients presented the chronic cardiac form (51.1%), followed by the indeterminate form (25.6%), and the cardiodigestive (23.2%) form at the time of the blood collection for *T. cruzi* culture (Table 2). Except for one patient with megacolon, all patients with the cardiodigestive form presented associated megaesophagus. No patient presented isolated digestive form. Among patients with the cardiac form, six presented the stage A, seven presented the stage B1, five presented the stage C, and four presented the stage D of the cardiac form. Among patients with the cardiodigestive form, five presented the stage A, two presented the stage B1, and three presented the stage C of the cardiac form.

### *T. cruzi* molecular typing

The DTUs identified in the blood culture isolates included only two subtypes: TcII in samples from 40 patients (93%) and TcVI in samples from 2 patients (4.6%). The molecular characterization was not possible to be done in the isolates obtained from one patient. The panel of the molecular targets identified in the *T. cruzi* isolates is described in Table 3.

Since DNA was extracted from the pellet of parasites from blood cultures, PCR products for all molecular targets were obtained in almost all samples. Representative images of agarose gels with amplifications for SL-IRac, SL-RI I and II, 24Sα r-DNA and A10 are shown in Fig 2. Regarding the two patients whose samples TcVI was identified, one was born in the city of Barreiras, located at the Western region of the state of Bahia, and the other was born in the city of Guimarânia, located at the state of Minas Gerais. TcII was identified in samples of patients born in Southeast and Northeast Brazilian regions and in the sample of the single case of the Midwest Brazilian region and from Bolivia (Fig 3 and Table 4).

**Table 2. Epidemiological characteristics of studied patients.**

| Variables | N = 43 | Percentage |
|---|---|---|
| **Sex** | | |
| Female | 31 | 72.1 |
| Male | 12 | 27.9 |
| Age (years) | 24–79 years (52.4±13.2) | |
| **Region of Origin** | | |
| Northeast | 33 | 76.7 |
| Southeast | 8 | 18.6 |
| Midwest | 1 | 2.3 |
| North | - | |
| South | - | |
| Bolivia | 1 | 2.3 |
| **Transmission Mode** | | |
| Vector borne | 39 | 90.7 |
| Congenital | 1 | 2.3 |
| Blood transfusion | 1 | 2.3 |
| Unknown | 2 | 4.6 |
| **Chagas disease clinical forms** | | |
| Cardiac | 22 | 51.1 |
| Indeterminate | 11 | 25.6 |
| Cardiodigestive | 10 | 23.3 |

## Clinical characteristics and follow-up according to *T. cruzi* DTUs

The mean follow-up time was 5.7 ± 4.4 years. A total of 14 patients (32.5%) died and one patient underwent heart transplantation. The cause of death was sudden cardiac arrest in six patients, heart failure in five patients, not related to Chagas disease in one patient, and ignored in two patients. A total of 8 patients (18.6%) with the cardiac or cardiodigestive forms progressed during the study follow-up: one from stage A to B2, two from stage A to C, two from stage B1 to B2, one from stage B1 to C, and two from stage C to D (Table 4).

The two patients with TcVI presented the cardiac form at the time of the blood culture was collected, one was at the stage A and another at the stage B1. The patient whose DTU was not determined presented the indeterminate form and did not present any event during the follow-up. All events (deaths, heart transplant, or Chagas disease clinical progression) occurred in patients with TcII. However, the low number of patients with TcVI precluded a comparison between outcomes of patients with TcII and TcVI.

A total of eight patients (18.6%) were treated with benznidazol during the study follow-up: one had the indeterminate form and seven had the cardiac form at the time they were treated.

## Discussion

The physiopathology of Chagas disease is still a mystery to be solved and the description of the different *T. cruzi* genotypes added a fundamental brick in the road towards the understanding of this disease. Therefore, the elucidation of which *T. cruzi* DTUs are associated to the different chronic Chagas disease presentations and their outcomes is necessary. In this article, we determined the DTU of parasites isolated by blood culture in 42 patients by means of multilocus conventional PCR. We found TcII to be the most common DTU among patients followed at our institution and that this DTU was associated both to patients with the indeterminate and

**Table 3. Panel of the molecular targets and DTUs identified in the *T. cruzi* isolates from blood cultures.**

| Samples | | Target Genes | | | | DTUs |
|---|---|---|---|---|---|---|
| Code | Number | SL-IRac | SL-IR I and II | 24Sα-rDNA | A10 | |
| EMT1 | 1175 | 157bp | 300bp | 140bp | 580bp | TcII |
| EMT2 a | 1176 | Neg | Neg | Neg | Neg | |
| EMT2 b | 1177 | 157bp | 300bp | 140bp | 580bp | TcII |
| EMT2 c | 1186 | 157bp | Neg | Neg | Neg | |
| EMT3 | 1190 | 157bp | 300bp | 140bp | 580bp | TcII |
| EMT4 a | 1178 | 157bp | 300bp | Neg | Neg | |
| EMT4 b | 1183 | Neg | 300bp | 140bp | 525bp | TcVI |
| EMT4 c | 1192 | 157bp | Neg | Neg | Neg | |
| EMT5 a | 1194 | 157bp | Neg | Neg | 580bp | TcII |
| EMT5 b | 1185 | Neg | 300bp | 140bp | Neg | |
| EMT6 a | 1191 | 157bp | Neg | Neg | Neg | |
| EMT6 b | 1243 | 157bp | 300bp | 140bp | 580bp | TcII |
| EMT7 a | 1196 | 157bp | 300bp | 140bp | Neg | |
| EMT7 b | 1197 | 157bp | Neg | Neg | Neg | |
| EMT7 c | 1220 | 157bp | 300bp | 140bp | 580bp | TcII |
| EMT8 | 1253 | 157bp | 300bp | 140bp | 580bp | TcII |
| EMT9 | 1273 | 157bp | 300bp | 140bp | 580bp | TcII |
| EMT10 | 1297 | 157bp | 300bp | 140bp | 580bp | TcII |
| EMT11 | 1198 | 157bp | 300bp | 140bp | 580bp | TcII |
| EMT12 | 1752 | 157bp | 300bp | 140bp | 580bp | TcII |
| EMT13 | 1290 | 157bp | 300bp | 140bp | 580bp | TcII |
| EMT14 | 1267 | 157bp | 300bp | 140bp | 580bp | TcII |
| EMT15 | 1282 | 157bp | 300bp | 140bp | 580bp | TcII |
| EMT16 | 1281 | 157bp | 300bp | 140bp | 580bp | TcII |
| EMT17 | 1274 | 157bp | 300bp | 140bp | 580bp | TcII |
| EMT18 | 1221 | 157bp | 300bp | 140bp | 580bp | TcII |
| EMT19 | 1283 | 157bp | 300bp | 140bp | 580bp | TcII |
| EMT20 | 1302 | 157bp | 300bp | 140bp | 580bp | TcII |
| EMT21 | 1319 | 157bp | 300bp | 140bp | 580bp | TcII |
| EMT22 | 1263 | 157bp | 300bp | 140bp | 580bp | TcII |
| EMT23 a | 1275A | Neg | 300bp | Neg | Neg | |
| EMT23 b | 1275B | 157bp | 300bp | 140bp | 580bp | TcII |
| EMT24 | 1199 | 157bp | 300bp | 140bp | 580bp | TcII |
| EMT25 | 1395 | 157bp | 300bp | 140bp | 580bp | TcII |
| EMT26 a | 1316 | 157bp | 300bp | 140bp | 580bp | TcII |
| EMT26 b | 1342 | 157bp | 300bp | Neg | Neg | |
| EMT27 | 1340 | 157bp | 300bp | 140bp | 580bp | TcII |
| EMT28 a | 1361 | 157bp | 300bp | Neg | Neg | |
| EMT28 b | 1362 | 157bp | 300bp | 140bp | 580bp | TcII |
| EMT29 a | 1395 | Neg | 300bp | Neg | 525bp | TcVI |
| EMT29 b | 1339 | 157bp | 300bp | 140bp | Neg | |
| EMT30 | 1363 | 157bp | 300bp | 140bp | 580bp | TcII |
| EMT31 a | 1364 | 157bp | 300bp | Neg | 580bp | TcII |
| EMT31 b | 1365 | 157bp | 300bp | Neg | Neg | |
| EMT32 a | 1396 | 157bp | 300bp | 140bp | 580bp | TcII |
| EMT32 b | 1397 | 157bp | Neg | Neg | Neg | |

*(Continued)*

**Table 3.** (Continued)

| Samples | | Target Genes | | | | DTUs |
|---|---|---|---|---|---|---|
| EMT33 | 1412 | 157bp | 300bp | 140bp | 580bp | TcII |
| EMT34 | 1411 | 157bp | 300bp | 140bp | 580bp | TcII |
| EMT35 | 1751 | 157bp | 300bp | 140bp | 580bp | TcII |
| EMT36 | 1749 | Neg | Neg | Neg | Neg | ND |
| EMT37 | 1754 | 157bp | 300bp | 140bp | 580bp | TcII |
| EMT38 | 1756 | 157bp | 300bp | 140bp | 580bp | TcII |
| EMT39 | 1750 | 157bp | 300bp | 140bp | 580bp | TcII |
| EMT40 | 1759 | 157bp | 300bp | 140bp | 580bp | TcII |
| EMT41 | 1755 | 157bp | 300bp | 140bp | 580bp | TcII |
| EMT42 | 1752 | 157bp | 300bp | 140bp | 580bp | TcII |
| EMT43 | 1265 | 157bp | 300bp | 140bp | 580bp | TcII |

bp, base pairs; Neg, negative; ND, not detected.

cardiac forms and to patients that evolved with heart failure or sudden cardiac arrest, the two most common and ominous consequences of the cardiac form of Chagas disease.

The epidemiological characteristics of the studied sample of the present article were representative of the group of patients followed at the outpatient clinic of our institution, as a recent paper of our group that included 619 Chagas disease patients found similar clinical and epidemiological characteristics, including an elevated mean age and women predominance [34]. The urban cohorts are usually composed by patients that migrated from rural areas of the states of Bahia, Pernambuco, Paraíba, and Minas Gerais [34, 35]. Regarding to clinical forms, there was a predominance of the cardiac form, associated or not to digestive complications

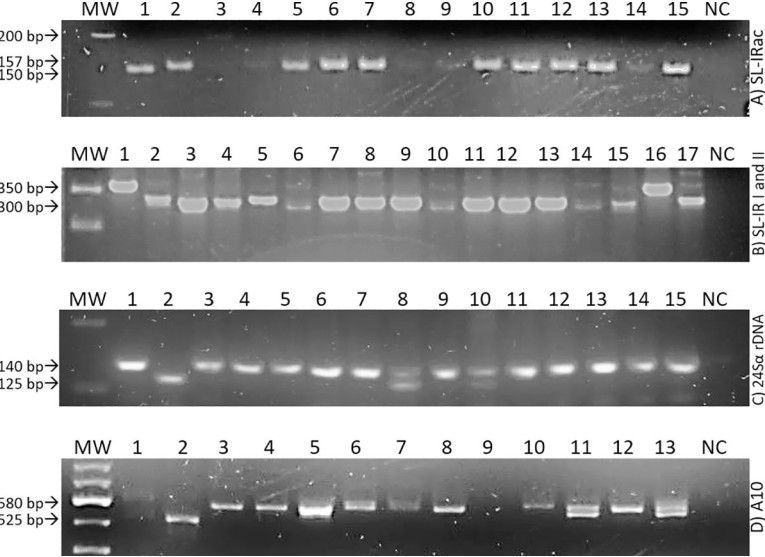

**Fig 2. Representative agarose gels showing *T. cruzi* target genes amplified by multilocus conventional PCR.** A) SL-IRac target (150/157bp/200bp). Lanes: 1- TcI (Dm28c), 2- TcII (Y), 3- TcIV (INPA4167), 4–15 –Patient samples; B) SL-IR I and II target (300/350bp). Lanes: 1- TcI (Dm28c), 2- TcII (Y), 3–15 –Patient samples, 16- TcI (Dm28c), 17- TcII (Y);, C) 24Sα rDNA target (125/140bp). Lanes: 1- TcII (Y), 2- TcIII (INPA3663), 3–15 –Patient samples; D) A10 target (525/580bp). Lanes: 1- TcII (Y), 2- TcVI (CL), 3–13 –Patient samples. MW, molecular weight; NC, negative control; bp, base pairs.

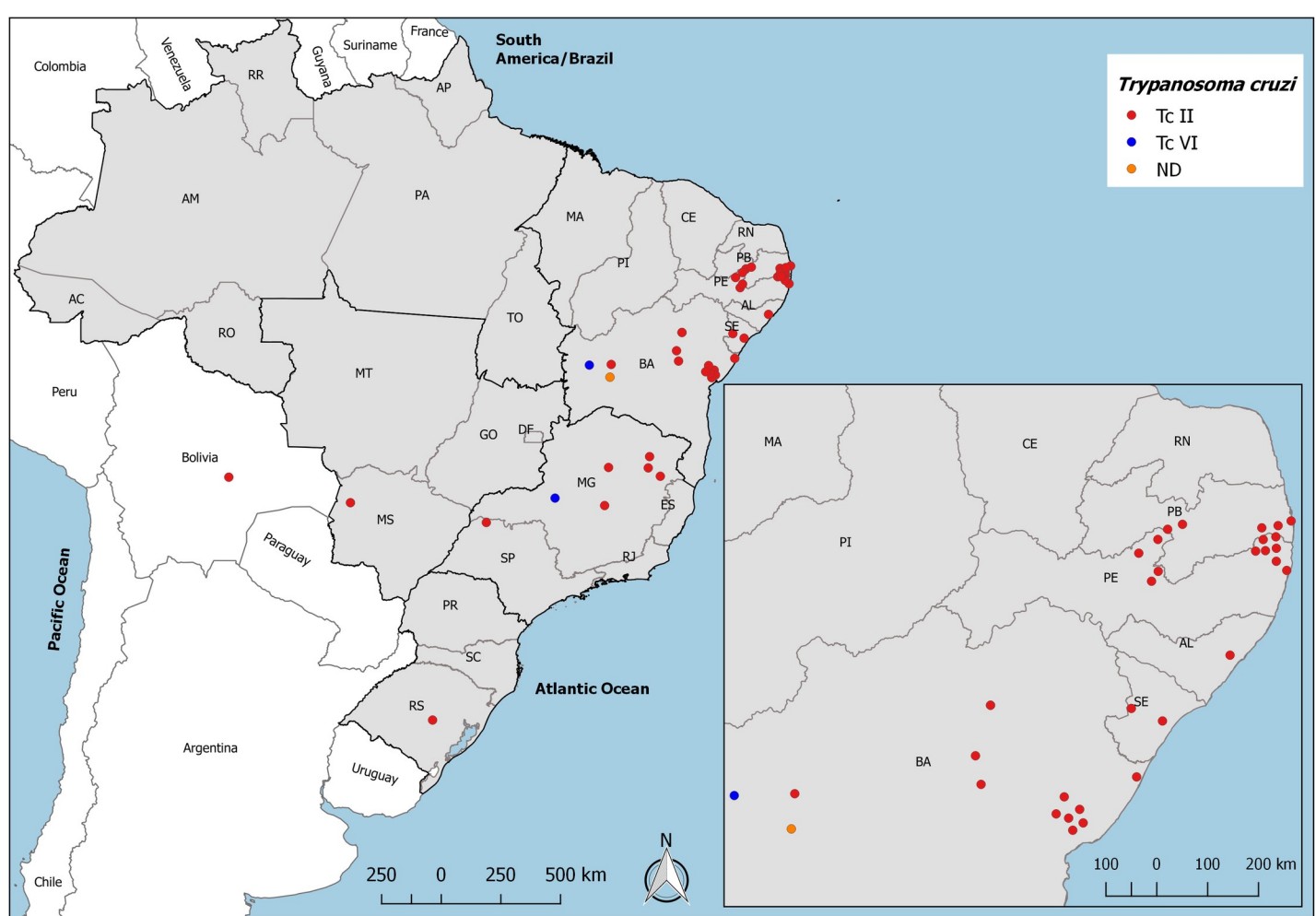

**Fig 3. Spatial distribution of *T. cruzi* DTUs from INI cohort Chagas disease patients according to their place of birth, except for the case of congenital transmission that was located according to his mother place of birth (Cachoeira do Sul, RS).** This map was created using QGIS version 3.4 software and cartographic bases maps modified from open access by the Brazilian Institute of Geography and Statistics, IBGE (https://www.ibge.gov.br/geociencias/downloads-geociencias.html). ND, not detected.

(cardiodigestive form), similar to the findings of other studies [34, 36]. The predominance of patients with cardiac form in this study can be attributed to two facts: the Chagas disease referral role of our institution and the higher adherence to treatment and follow-up among symptomatic patients than among patients with the indeterminate form. Other group of patients that compose our cohort are those referred by blood banks, who are usually younger and asymptomatic [37]. The most common mode of transmission was vector borne, which reflects the pattern of migration from rural area that compose our cohort.

The predominance of the TcII DTU is in accordance to other studies that enrolled patients from rural endemic areas from the Northeast and Southeast Brazilian regions where domiciliary vector borne is the main mode of transmission [11, 13, 14, 23]. In fact, TcII is the most found DTU in genotyping studies performed in Brazil and is associated to severe chronic cardiac forms and chronic digestive forms [7, 14, 23]. The TcII and TcVI were found in patients with the chronic cardiac form of Chagas disease in Brazil and in other South American countries, such as Argentina, Bolivia, and Chile [11, 14, 15, 22, 23, 38, 39]. However, most *T. cruzi* genotyping studies focused on the association between specific DTU genotypes and the cardiac

**Table 4. Clinical, epidemiological, follow-up, outcome, and DTU classification of studied patients.**

| Case | Geographic Origin (City/State) | Age | Clinical Form (stage) | Follow-up Time (years) | Progression | Death | DTU |
|------|-------------------------------|-----|----------------------|------------------------|-------------|-------|-----|
| 1 | Barreiras/BA | 26 | Cardiac (B1) | 11.5 | No | No | TcVI |
| 2 | Cachoeira/BA | 79 | Cardiodigestive (stage C+ME II) | 1.73 | Yes (stage D) | Yes | TcII |
| 3 | Cachoeira/BA | 77 | Cardiac (A) | 0.29 | No | No | TcII |
| 4 | Cachoeira/BA | 64 | Cardiodigestive (stage A+ME I) | 10.41 | Yes (stage C) | No | TcII |
| 5 | Campo Formoso/BA | 35 | Cardiodigestive (stage A+ME I) | 3.6 | No | No | TcII |
| 6 | Campo Formoso/BA | 49 | Cardiac (B1) | 9.44 | No | No | TcII |
| 7 | Conde/BA | 64 | Cardiodigestive (stage A+MC) | 11.3 | No | No | TcII |
| 8 | Feira de Santana/BA | 58 | Cardiodigestive (stage A+ME II) | 7.92 | Yes (stage C) | Yes | TcII |
| 9 | Miguel Calmon/BA | 61 | Cardiac (D) | 7.73 | No | Yes | TcII |
| 10 | Mundo Novo/BA | 56 | Cardiodigestive (stage A+ME I) | 0.61 | No | No | TcII |
| 11 | São Félix/BA | 63 | Cardiodigestive (stage B1+ME II) | 10.76 | No | No | TcII |
| 12 | São Francisco do Conde/BA | 56 | Cardiac (A) | 11.33 | No | No | TcII |
| 13 | Serra Dourada/BA | 31 | Indeterminate | 9.48 | No | No | ND |
| 14 | Wanderley/BA | 36 | Indeterminate | 0.1 | No | No | TcII |
| 15 | Afogados da Ingazeira/PE | 39 | Indeterminate | 9.56 | No | No | TcII |
| 16 | Aliança/PE | 71 | Cardiodigestive (stage B1+MEIV) | 10.96 | No | No | TcII |
| 17 | Araçoiaba/PE | 50 | Cardiac (A) | 11.25 | Yes (stage B2) | No | TcII |
| 18 | Itambé/PE | 60 | Cardiac (B1) | 4.88 | No | Yes | TcII |
| 19 | Machados/PE | 63 | Cardiac (A) | 0.57 | No | No | TcII |
| 20 | Recife/PE | 42 | Indeterminate | 0.85 | No | No | TcII |
| 21 | São José do Egito/PE | 54 | Cardiodigestive (stage C+ME I) | 10.09 | No | No | TcII |
| 22 | Sertânia/PE | 45 | Indeterminate | 9.46 | No | No | TcII |
| 23 | Sertânia/PE | 40 | Indeterminate | 9.12 | No | No | TcII |
| 24 | Timbaúba/PE | 59 | Cardiac (B1) | 4.13 | Yes (stage C) | Yes | TcII |
| 25 | Timbaúba/PE | 61 | Cardiac (B1) | 9.03 | Yes (stage B2) | No | TcII |
| 26 | Araçuaí/MG | 56 | Cardiac (B1) | 7.34 | Yes (stage B2) | Yes | TcII |
| 27 | Engenheiro Navarro/MG | 79 | Cardiodigestive (stage C+MEIII) | 0.44 | No | Yes | TcII |
| 28 | Guimarânia/MG | 64 | Cardiac (A) | 5.17 | No | No | TcVI |
| 29 | Novo Cruzeiro/MG | 55 | Cardiac (C) | 2.15 | No | Yes | TcII |
| 30 | Novo Cruzeiro/MG | 51 | Cardiac (C) | 9.58 | No | No | TcII |
| 31 | Teófilo Otoni/MG | 52 | Cardiac (C) | 9.41 | Yes (stage D) | Yes | TcII |
| 32 | Desterro/PB | 37 | Cardiac (C) | 6.82 | No | Yes | TcII |
| 33 | Itabaiana/PB | 62 | Cardiac (A) | 0.22 | No | No | TcII |
| 34 | João Pessoa/PB | 52 | Cardiac (C) | 2.92 | No | Yes | TcII |
| 35 | Pedras de Fogo/PB | 53 | Cardiac (D) | 2.36 | No | Yes | TcII |
| 36 | Taperoá/PB | 42 | Cardiac (D) | 1.27 | No | Heart Transplant | TcII |
| 37 | Laranjeiras/SE | 47 | Cardiac (D) | 0.13 | No | Yes | TcII |
| 38 | Pinhão/SE | 62 | Cardiac (B1) | 1.21 | No | Yes | TcII |
| 39 | Pilar/AL | 36 | Indeterminate | 11.45 | No | No | TcII |
| 40 | Nova Iguaçu/RJ | 52 | Indeterminate | 10.88 | No | No | TcII |
| 41 | Macedônia/SP | 45 | Indeterminate | 0.24 | No | No | TcII |
| 42 | Corumbá/MS | 24 | Indeterminate | 0.88 | No | No | TcII |
| 43 | Santa Cruz de La Sierra/Bolivia | 46 | Indeterminate | 0.08 | No | No | TcII |

Cardiac form stages: A, B1, B2, C, D; MC: megacolon; ME: megaesophagus; ME stages: I, II, III, IV.

form of the disease and there are few studies that addressed such associations with the digestive form of the disease. In our study, one quarter of the patients presented the cardiodigestive form, which points to the association of TcII and digestive complications of Chagas disease. The prevalence of the cardiodigestive form in our studied sample is higher than expected [3]. On the other hand, in Argentina, TcV and TcVI were significantly associated with the digestive form of Chagas disease [24].

Our study showed that infections by TcII in Brazil can be associated to several types of chronic Chagas disease clinical presentations, including indeterminate, cardiac and cardiodigestive forms. Other study not only confirmed the association of TcII with different chronic Chagas disease forms, but also indicated that genetic variability within TcII is not associated with a specific clinical manifestation [23]. Our study also showed that important Chagas disease complications, such as heart failure and sudden cardiac arrest, and a high mortality occur in patients infected by TcII. All progressions that we observed occurred within the cardiac form, from an initial stage to a more advanced one. No patient with the indeterminate form progressed to the cardiac form during the study period. However, Chagas disease progression rate from indeterminate to the cardiac form of our cohort is low, 1.48 cases/100 patient years [35], and probably the low number of patients with the indeterminate form and their relatively short period of follow-up in the present study is not appropriated to address Chagas disease progression.

On the other hand, both patients of the present study infected by TcVI *T. cruzi* genotype presented the cardiac form and none of them progressed or died. However, the number of TcVI patients is too low for any conclusion to be drawn. Other studies that enrolled patients of our institution also identified TcVI genetic material in the blood drawn from patients born in the Southeast and Northeast Brazilian regions [11, 40]. TcVI was found to be associated to both cardiac and indeterminate forms of chronic Chagas disease in Brazil [11, 40, 41]. It is highly likely that immunologic factors of the host together with parasite genetic variations contribute to the diversity of Chagas disease clinical presentation [6]. Patients with the cardiac form present an inflammatory profile that may be related to disease progression [42]. For instance, serum TNF levels are reported to be higher among patients with the chronic cardiac form than in patients with the indeterminate form [43, 44].

Although a few reports have shown that the parasite population infecting specific organs can be genetically distinct from the population found in the patients' blood [29, 45, 46], it is unlikely that this occurs in the cohort of patients here analyzed due to the high prevalence of TcII found by us, which is in accordance to other studies that enrolled Brazilian patients [11, 13, 14, 23].

The analysis of the DTUs spatial distribution revealed that TcII is largely distributed in endemic Brazilian areas from Northeast, and Southeast regions, with a higher concentration in the East region of the states of Bahia, Paraíba, Pernambuco, Alagoas, and Sergipe. Patients infected with TcII were also born in semiarid areas of Northeastern states, mainly Bahia, Pernambuco, and Paraíba. In the Southeast region, patients infected with TcII were born mainly in the mid-north area of Minas Gerais state, but were also located in the West of the state of São Paulo. This TcII spatial distribution is in accordance with previous findings of other studies [11, 13, 14, 40]. There was only one case of TcII in the Mid-West region which limits our discussion. The case from Bolivia was identified as TcII and, although this is not the most frequently DTU identified in that country, other studies confirmed the presence of TcII in Bolivia [17, 22]. One case of congenital transmission was located in the South region, municipality of Cachoeira do Sul, as that was the place of birth of the mother of this case.

The two TcVI cases were original to central Brazilian areas, West of the states of Bahia and Minas Gerais. *T. cruzi* TcVI had already been identified in the same TcII spatial distribution, either as an isolated infection or as a mixed infection (TcII/TcVI) [11, 13, 40, 41].

## Study limitations

Although TcII is also associated to patients with chronic digestive form [10, 22, 23, 40], in our sample we did not identify any patient with the digestive form among those with TcII. This is possibly due to the low prevalence of patients with the digestive form in our cohort, around 5% [34], and the small sample of the present study. Therefore, we could not evaluate association between DTU genotypes and chronic digestive form.

Only TcII and TcVI were detected among the studied patients. The DTU identification was not possible in one of the studied patients as none of the PCR targets could be amplified in any of the two samples of this case. We believe that this occurred due to parasite degradation during the storage period or the presence of PCR inhibitors or DNA loss in this sample. Also, patients' samples were not genotyped at more than one time point during their follow-up and we could not analyze if there were changes in parasite DTU genotypes over time.

The absence of mixed infections in the present paper may be due to the blood culture method that could select a specific *T. cruzi* subpopulation more adapted to the selected medium, as previously observed [14, 24, 38]. New approaches based on deep sequencing could have overcome this limitation as this technique was able to identify in a group of 17 patients from Mexico, 8 patients (47%) harboring infections with multiple DTUs [47]. However, the multilocus PCR used by us is also able to identify mixed infections directly from patient's blood and is recommended by experts' consensus statement to be used in clinical samples [9].

## Conclusions

The TcII was the main *T. cruzi* DTU genotype isolated from blood culture samples obtained from chronic Chagas disease Brazilian patients with either cardiac, indeterminate or cardiodigestive forms born at Southeast and Northeast regions. Other DTU genotype found in much less frequency was TcVI. *T. cruzi* TcII was also associated to patients that evolved with heart failure or sudden cardiac arrest, the two most common and ominous consequences of the cardiac form of Chagas disease.

## Supporting information

**S1 Table. Patients list with positive blood cultures samples, age, gender, transmission mode, serological results, DTUs, geographical origin and coordinates.**
(DOCX)

**S1 Raw images.**
(PDF)

## Author Contributions

**Conceptualization:** Luiz Henrique Conde Sangenis, Roberto Magalhães Saraiva.

**Data curation:** Luciana de Freitas Campos Miranda, Alejandro Marcel Hasslocher-Moreno, Roberto Magalhães Saraiva.

**Formal analysis:** Otacílio C. Moreira, Roberto Magalhães Saraiva.

**Funding acquisition:** Roberto Magalhães Saraiva.

**Investigation:** Marco Antonio Prates Nielebock, Samanta Cristina das Chagas Xavier, Luciana de Freitas Campos Miranda, Ana Carolina Bastos de Lima, Thayanne Oliveira de Jesus Sales Pereira, Luiz Henrique Conde Sangenis.

**Methodology:** Marco Antonio Prates Nielebock, Otacílio C. Moreira, Samanta Cristina das Chagas Xavier, Luciana de Freitas Campos Miranda, Ana Carolina Bastos de Lima, Constança Britto, Luiz Henrique Conde Sangenis, Roberto Magalhães Saraiva.

**Project administration:** Otacílio C. Moreira, Roberto Magalhães Saraiva.

**Supervision:** Roberto Magalhães Saraiva.

**Visualization:** Constança Britto.

**Writing – original draft:** Marco Antonio Prates Nielebock, Otacílio C. Moreira, Luiz Henrique Conde Sangenis, Roberto Magalhães Saraiva.

**Writing – review & editing:** Marco Antonio Prates Nielebock, Otacílio C. Moreira, Samanta Cristina das Chagas Xavier, Luciana de Freitas Campos Miranda, Ana Carolina Bastos de Lima, Thayanne Oliveira de Jesus Sales Pereira, Alejandro Marcel Hasslocher-Moreno, Constança Britto, Luiz Henrique Conde Sangenis, Roberto Magalhães Saraiva.

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
