## [Decision Letter · Decision Letter 0]

6 Oct 2020

PONE-D-20-26324

Association between Trypanosoma cruzi genotypes and chronic Chagas disease clinical presentation and outcome in an urban cohort in Brazil.

PLOS ONE

Dear Dr. Roberto M Saraiva:

Thank you for submitting your manuscript to PLOS ONE. After careful consideration, we feel that it has merit but does not fully meet PLOS ONE’s publication criteria as it currently stands. Therefore, we invite you to submit a revised version of the manuscript that addresses the points raised during the review process.

We look forward to receiving your revised manuscript.

Kind regards,

Claudia Patricia Herrera, Ph.D.

Academic Editor

PLOS ONE

Journal Requirements:

2. In your Methods section, please provide additional information about the participant recruitment method and the demographic details of your participants. Please ensure you have provided sufficient details to replicate the analyses such as: a) the recruitment date range (month and year),  b) a description of how participants were recruited, and c) descriptions of where participants were recruited and where the research took place. Please also describe the methods used to collect patient samples.

4. Thank you for including your ethics statement:

"This study was approved by the Institutional Review Bord under number 62973116.6.0000.5262. All procedures followed regulatory guidelines and standards for research involving human beings as stated in the Brazilian National Health Council Resolution 466/2012 and were conducted according to the principles expressed in the Declaration of Helsinki in order to safeguard the rights and welfare of the participants. ".

i) Please amend your current ethics statement to include the full name of the ethics committee/institutional review board(s) that approved your specific study.

ii) Once you have amended this/these statement(s) in the Methods section of the manuscript, please add the same text to the “Ethics Statement” field of the submission form (via “Edit Submission”).

5. Please provide additional details regarding participant consent. In the ethics statement in the Methods and online submission information, please ensure that you have specified (1) whether consent was informed and (2) what type you obtained (for instance, written or verbal, and if verbal, how it was documented and witnessed). If your study included minors, state whether you obtained consent from parents or guardians. If the need for consent was waived by the ethics committee, please include this information.

7. We note that Figure 3 in your submission contain map images which may be copyrighted. All PLOS content is published under the Creative Commons Attribution License (CC BY 4.0), which means that the manuscript, images, and Supporting Information files will be freely available online, and any third party is permitted to access, download, copy, distribute, and use these materials in any way, even commercially, with proper attribution. For these reasons, we cannot publish previously copyrighted maps or satellite images created using proprietary data, such as Google software (Google Maps, Street View, and Earth). For more information, see our copyright guidelines: http://journals.plos.org/plosone/s/licenses-and-copyright.

7.1.    You may seek permission from the original copyright holder of Figure 3 to publish the content specifically under the CC BY 4.0 license. 

7.2.    If you are unable to obtain permission from the original copyright holder to publish these figures under the CC BY 4.0 license or if the copyright holder’s requirements are incompatible with the CC BY 4.0 license, please either i) remove the figure or ii) supply a replacement figure that complies with the CC BY 4.0 license. Please check copyright information on all replacement figures and update the figure caption with source information. If applicable, please specify in the figure caption text when a figure is similar but not identical to the original image and is therefore for illustrative purposes only.

Reviewers' comments:

Reviewer's Responses to Questions

**Comments to the Author**

1. Is the manuscript technically sound, and do the data support the conclusions?

Reviewer #1: Yes

Reviewer #2: Yes

2. Has the statistical analysis been performed appropriately and rigorously? 

Reviewer #1: Yes

Reviewer #2: N/A

3. Have the authors made all data underlying the findings in their manuscript fully available?

Reviewer #1: Yes

Reviewer #2: Yes

4. Is the manuscript presented in an intelligible fashion and written in standard English?

Reviewer #1: Yes

Reviewer #2: Yes

5. Review Comments to the Author

Reviewer #1: This is an interesting study that sought to evaluate a possible association between the infective T. cruzi genotype and the clinical manifestation of Chagas disease in Brazilian patients. Although the results confirm data from other authors that TcII is prevalent in patients from various regions of Brazil and that DTU TcII promotes the indeterminate, cardiac and cardiodigestive forms in the chronic phase, the study deserves to be published not only because it expands knowledge on the subject, but also because it is a well-conducted study that also analyzes the progression of the disease in some patients.

To supplement the MS, this reviewer raises some relevant points that must be taken into account before the final acceptance. These points are listed below:

- Title: Association between Trypanosoma cruzi genotypes and chronic Chagas disease clinical presentation and outcome in an urban cohort in Brazil.

The title is very assertive. The reader is induced to think that the authors found this association, when, in fact, TcII is responsible for cardiac, indeterminate and cardiodigestive forms. Therefore, the following title is suggested:

Association between Trypanosoma cruzi DTU TcII and chronic Chagas disease clinical presentations and outcome in an urban cohort in Brazil.

Background

Line 32: Substitution: This study aimed to identify the potential association between T. cruzi genotypes and the clinical presentations of chronic Chagas disease.

Methodology/Principal Findings

Suggestion: Start by describing the characteristics of the patients. Then, describe the DTU typing method and the data obtained.

Conclusions/Significance

Provide percentages:

Line 49 - TcII was the main T. cruzi DTU identified in chronic Chagas disease Brazilian patients (45.2%)….

Line 51 - Other DTU found in much less frequency (4.7%) was TcVI.

Introduction

Line 82 -…… Latin America, mainly in Argentina, Brazil, Mexico, Bolivia and Colombia [2,3].

Why are countries mentioned in this order? Is it in decreasing order of prevalence? If not, quote in alphabetical order: Argentina, Bolivia, Brazil, Colombia and Mexico

Line 101 – Reassess the sentence: “Therefore, TcII, TcV, and TcVI are the T. cruzi genotypes with the highest pathogenic potential as they are related both to cardiac and digestive chronic clinical Chagas disease forms [6]”.

This reviewer does not agree with the statement that these DTUs have the highest pathogenic potential BECAUSE they cause cardiac and digestive forms. TcI is also highly pathogenic and causes severe cardiomyopathy in chronic chagasic patients in Argentina (Burgos et al. Clin Infect Dis. 2010; 51: 485–495) and Colombia (Ramírez et al. PLoS Negl Trop Dis. 2010 doi: 10.1371 / journal.pntd.0000899) as well as Venezuela.

Methods. Patients and Study Design:

Briefly inform the tests used for clinical evaluation of the cardiac, digestive and cardiodigestive forms.

Results

Line 208 – Inform percentage: Most patients were women (72.2%) and were infected by vector borne transmission (Table 2).

Line 260 – Rephrase: A total of 8 patients (18.6%) with the cardiac or cardiodigestive forms progressed during the study follow-up,

Discussion

Some relevant aspects should be included in the Discussion to make it more comprehensive and hypotheses should be raised by the authors to explain some observations.

- In the cohort, there were no patients with only the digestive form. On the other hand, 23% of the patients had the cardiodigestive form. Since several authors show that TcII promotes the digestive form (cite references). To what factors do the authors attribute the findings of the present study? (see Lines 322-331).

- Comment on a possible criticism:

Although a few reports have shown that the parasite population infecting specific organs can be genetically distinct from the population found in the patients’ blood (cite references), it is unlikely that this occurs in the cohort of patients here analyzed.

- A clear conclusion of the study is that TcII promotes different clinical presentations of ChD. The studies by Lages-Silva et al. (Ref. 23), which indicate that the genetic variability within TcII is not associated with the clinical manifestation, should be cited.

- On Line 346 the authors mention that “immunologic factors of the host together with parasite genetic variation contribute to the diversity of Chagas disease clinical presentation [6]”. Authors should further develop this hypothesis by presenting some studies that can support it.

The sentences below can be improved

Line 282 Therefore, it is needed to describe which T. cruzi DTUs are found in patients with Chagas disease in Brazil and which clinical presentations and outcomes can be associated to those DTUs.

Line 299 Our (?) predominance of patients with cardiac form can

Move Fig. 2 to Supplemental data.

Reviewer #2: This study aims at identifying T. cruzi parasite genotypes and correlate these with clinical manifestations in a convenience sample of patients. It is an important topic and the study is overall well performed and well presented. The major limitation is the methodological approach used, which is not the most appropriate for such studies. Indeed, genotyping by PCR from isolated parasites generates bias as in vitro culture selects for strains growing faster in the selected medium, and genotyping by PCR only detects the dominant strains in potential mixtures of strains. More sensitive approaches based on direct genotyping by sequencing, and particularly deep sequencing, have shown that infections with multiple strains/DTUs can be frequent in some regions (see for example Villanueva-Lizama et al., J Infect Dis, 2019. 219(12): 1980-1988) and may have been overlooked in this study. These aspects should at least be discussed.

Minor comments:

Follow-up time should be indicated for each patient in Table 4. Also, were any patient’s samples genotyped at more than 1 time point during their follow-up? If yes, were there any changes in parasite genotypes over time as seen in some studies? Has parasitemia been quantified in these patients? Were the patients treated? If yes, how did they respond to treatment? These aspects would enrich the study.

6. PLOS authors have the option to publish the peer review history of their article (what does this mean?). If published, this will include your full peer review and any attached files.

Reviewer #1: No

Reviewer #2: **Yes: **Eric Dumonteil

---

## [Author Response · Author response to Decision Letter 0]

30 Oct 2020

Question: If applicable, we recommend that you deposit your laboratory protocols in protocols.io to enhance the reproducibility of your results. Protocols.io assigns your protocol its own identifier (DOI) so that it can be cited independently in the future. For instructions see: http://journals.plos.org/plosone/s/submission-guidelines#loc-laboratory-protocols

Answer: In the original manuscript, all protocols used are described in the cited references in sufficient details to allow replication of experiments. Please, see page 5, line 139; page 6, line 160; and references 11, 19, 26, and 29 of the revised manuscript.

Journal Requirements:

Question: 1. Please ensure that your manuscript meets PLOS ONE's style requirements, including those for file naming. The PLOS ONE style templates can be found at

Answer: All changes necessary for our manuscript to meet PLOS ONE’s style requirements were done.

Question: 2. In your Methods section, please provide additional information about the participant recruitment method and the demographic details of your participants. Please ensure you have provided sufficient details to replicate the analyses such as: a) the recruitment date range (month and year), b) a description of how participants were recruited, and c) descriptions of where participants were recruited and where the research took place. Please also describe the methods used to collect patient samples.

Answer: The research took place at the Evandro Chagas National Institute of Infectious Diseases, Fiocruz, Rio de Janeiro, Brazil. This was a retrospective study based on parasites isolated from blood cultures collected between July 2008 and June 2010. Participants were approached during their regular medical appointment at our outpatient clinic in order to give written informed consent to allow the use of the archived parasites isolates. The demographic details of the study participants are described in Table 2. All methods are described in detail. Please see page 4, lines 107 to 112 for changes in the revised manuscript.

Question: 3. We suggest you thoroughly copyedit your manuscript for language usage, spelling, and grammar. If you do not know anyone who can help you do this, you may wish to consider employing a professional scientific editing service. 

Answer: We have copyedited the revised version of our manuscript.

Question: 4. Thank you for including your ethics statement:

"This study was approved by the Institutional Review Bord under number 62973116.6.0000.5262. All procedures followed regulatory guidelines and standards for research involving human beings as stated in the Brazilian National Health Council Resolution 466/2012 and were conducted according to the principles expressed in the Declaration of Helsinki in order to safeguard the rights and welfare of the participants. ".

i) Please amend your current ethics statement to include the full name of the ethics committee/institutional review board(s) that approved your specific study.

Answer: We have included the full name of the ethics committee. Please, see page 5, lines 128 and 129 of the revised manuscript.

Question: ii) Once you have amended this/these statement(s) in the Methods section of the manuscript, please add the same text to the “Ethics Statement” field of the submission form (via “Edit Submission”).

Answer:We made the requested correction.

Question: 5. Please provide additional details regarding participant consent. In the ethics statement in the Methods and online submission information, please ensure that you have specified (1) whether consent was informed and (2) what type you obtained (for instance, written or verbal, and if verbal, how it was documented and witnessed). If your study included minors, state whether you obtained consent from parents or guardians. If the need for consent was waived by the ethics committee, please include this information.

Answer:This is a retrospective study using archived samples. All participants who were still followed at our institution provided written informed consent allowing the use of their archived parasites isolated from their blood culture and granting access to their medical records. The institutional ethics committee waived the requirement for informed consent for deceased participants and those who were lost to follow-up and could not be reached. Please see page 4, lines 107 to 112 for changes in the revised manuscript.

Question: 6. PLOS ONE now requires that authors provide the original uncropped and unadjusted images underlying all blot or gel results reported in a submission’s figures or Supporting Information files. This policy and the journal’s other requirements for blot/gel reporting and figure preparation are described in detail at https://journals.plos.org/plosone/s/figures#loc-blot-and-gel-reporting-requirements and https://journals.plos.org/plosone/s/figures#loc-preparing-figures-from-image-files. When you submit your revised manuscript, please ensure that your figures adhere fully to these guidelines and provide the original underlying images for all blot or gel data reported in your submission. See the following link for instructions on providing the original image data: https://journals.plos.org/plosone/s/figures#loc-original-images-for-blots-and-gels.

Ansewer: All original uncropped and unadjusted gel images used in Figure 2 were uploaded as Supporting Information file named S1_raw_images as a pdf file. 

Question: 7. We note that Figure 3 in your submission contain map images which may be copyrighted. All PLOS content is published under the Creative Commons Attribution License (CC BY 4.0), which means that the manuscript, images, and Supporting Information files will be freely available online, and any third party is permitted to access, download, copy, distribute, and use these materials in any way, even commercially, with proper attribution. For these reasons, we cannot publish previously copyrighted maps or satellite images created using proprietary data, such as Google software (Google Maps, Street View, and Earth). For more information, see our copyright guidelines: http://journals.plos.org/plosone/s/licenses-and-copyright.

Answer: The map presented in Figure 3 was created using cartographic bases maps modified from open access (public domain) by the Brazilian Institute of Geography and Statistics, IBGE (https://www.ibge.gov.br/geociencias/downloads-geociencias.html). Therefore, the maps used are not copyrighted. This information was added to the Methods section and Figure 3 legend. Please see page 8, lines 208 to 210; and page 13 lines 310 to 313 for changes in the revised manuscript.

Question: 8. Please include captions for your Supporting Information files at the end of your manuscript, and update any in-text citations to match accordingly. Please see our Supporting Information guidelines for more information: http://journals.plos.org/plosone/s/supporting-information.

Answer: The captions for the supporting information files were included at the end of the revised manuscript. Please, see page 31 of the revised manuscript. 

Reviewers' comments:

Comments to the Author

Reviewer #1: This is an interesting study that sought to evaluate a possible association between the infective T. cruzi genotype and the clinical manifestation of Chagas disease in Brazilian patients. Although the results confirm data from other authors that TcII is prevalent in patients from various regions of Brazil and that DTU TcII promotes the indeterminate, cardiac and cardiodigestive forms in the chronic phase, the study deserves to be published not only because it expands knowledge on the subject, but also because it is a well-conducted study that also analyzes the progression of the disease in some patients.

To supplement the MS, this reviewer raises some relevant points that must be taken into account before the final acceptance. These points are listed below:

Question:- Title: Association between Trypanosoma cruzi genotypes and chronic Chagas disease clinical presentation and outcome in an urban cohort in Brazil.

The title is very assertive. The reader is induced to think that the authors found this association, when, in fact, TcII is responsible for cardiac, indeterminate and cardiodigestive forms. Therefore, the following title is suggested:

Association between Trypanosoma cruzi DTU TcII and chronic Chagas disease clinical presentations and outcome in an urban cohort in Brazil.

Answer: The title was changed according to the reviewer’s suggestion. 

Background

Question: Line 32: Substitution: This study aimed to identify the potential association between T. cruzi genotypes and the clinical presentations of chronic Chagas disease.

Answer: The text was changed according to the reviewer’s suggestion. Please, see page2, line 33 of the revised manuscript. 

Methodology/Principal Findings

Question: Suggestion: Start by describing the characteristics of the patients. Then, describe the DTU typing method and the data obtained.

Answer: The text was changed according to the reviewer’s suggestion. Please, see page 2, lines 36 to 41 of the revised manuscript. 

Question: Conclusions/Significance

Provide percentages:

Line 49 - TcII was the main T. cruzi DTU identified in chronic Chagas disease Brazilian patients (45.2%)….

Line 51 - Other DTU found in much less frequency (4.7%) was TcVI.

Answer:The percentages were provided. Please, see page 2, lines 51 and 52 of the revised manuscript. 

Question:Introduction

Line 82 -…… Latin America, mainly in Argentina, Brazil, Mexico, Bolivia and Colombia [2,3].

Why are countries mentioned in this order? Is it in decreasing order of prevalence? If not, quote in alphabetical order: Argentina, Bolivia, Brazil, Colombia and Mexico

Answer: The text was changed according to the reviewer’s suggestion. Please, see page 3, line 72 of the revised manuscript. 

Question: Line 101 – Reassess the sentence: “Therefore, TcII, TcV, and TcVI are the T. cruzi genotypes with the highest pathogenic potential as they are related both to cardiac and digestive chronic clinical Chagas disease forms [6]”.

This reviewer does not agree with the statement that these DTUs have the highest pathogenic potential BECAUSE they cause cardiac and digestive forms. TcI is also highly pathogenic and causes severe cardiomyopathy in chronic chagasic patients in Argentina (Burgos et al. Clin Infect Dis. 2010; 51: 485–495) and Colombia (Ramírez et al. PLoS Negl Trop Dis. 2010 doi: 10.1371 / journal.pntd.0000899) as well as Venezuela.

Answer: The sentence was excluded. 

Question: Methods. Patients and Study Design:

Briefly inform the tests used for clinical evaluation of the cardiac, digestive and cardiodigestive forms.

Answer: The tests used to evaluate the cardiac and digestive forms were included in the Methods section. Please, see page 5, lines 123 to 125 of the revised manuscript. 

Question: Results

Line 208 – Inform percentage: Most patients were women (72.2%) and were infected by vector borne transmission (Table 2).

Answer: The percentage were included in the text. Please, see page 9, line 237 of the revised manuscript. 

Question: Line 260 – Rephrase: A total of 8 patients (18.6%) with the cardiac or cardiodigestive forms progressed during the study follow-up,

Answer: The text was changed according to the reviewer’s suggestion. Please, see page 15, lines 345 to 348 of the revised manuscript. 

Question: Discussion

Some relevant aspects should be included in the Discussion to make it more comprehensive and hypotheses should be raised by the authors to explain some observations.

- In the cohort, there were no patients with only the digestive form. On the other hand, 23% of the patients had the cardiodigestive form. Since several authors show that TcII promotes the digestive form (cite references). To what factors do the authors attribute the findings of the present study? (see Lines 322-331).

Answer: The prevalence of patients with only the digestive form is low in our cohort: around 5% [34]. As the studied sample was relatively small, the absence of patients with only the digestive form was probably due by chance and the we cannot evaluate the association between DTU and the digestive form in the present study. Please, see page 19, lines 479 to 484 for changes in the revised manuscript. 

Question: - Comment on a possible criticism:

Although a few reports have shown that the parasite population infecting specific organs can be genetically distinct from the population found in the patients’ blood (cite references), it is unlikely that this occurs in the cohort of patients here analyzed.

Answer: The comment suggested by the reviewer was added to the text. Please, see page 18, lines 454 to 458 of the revised manuscript. 

Question: - A clear conclusion of the study is that TcII promotes different clinical presentations of ChD. The studies by Lages-Silva et al. (Ref. 23), which indicate that the genetic variability within TcII is not associated with the clinical manifestation, should be cited.

Answer: The comment suggested by the reviewer was added to the text. Please, see page 17, lines 421 to 423 of the revised manuscript. 

Question: - On Line 346 the authors mention that “immunologic factors of the host together with parasite genetic variation contribute to the diversity of Chagas disease clinical presentation [6]”. Authors should further develop this hypothesis by presenting some studies that can support it.

Answer: We added references that support the hypothesis that immunology factors contribute to Chagas diseases different presentations. Please, see page 18, lines 450 to 453 of the revised manuscript. 

Question: The sentences below can be improved

Line 282 Therefore, it is needed to describe which T. cruzi DTUs are found in patients with Chagas disease in Brazil and which clinical presentations and outcomes can be associated to those DTUs.

Answer: The sentence was improved. Please, see page 15, line 363 to page 16, line 367 of the revised manuscript. 

Question Line 299 Our (?) predominance of patients with cardiac form can

Answer: The sentence was improved. Please, see page 16, line 373 of the revised manuscript. 

Question Move Fig. 2 to Supplemental data.

Answer: We highly appreciated all the reviewer’s comments but we prefer to keep Fig 2 in the main manuscript. 

Reviewer #2: This study aims at identifying T. cruzi parasite genotypes and correlate these with clinical manifestations in a convenience sample of patients. It is an important topic and the study is overall well performed and well presented. The major limitation is the methodological approach used, which is not the most appropriate for such studies. Indeed, genotyping by PCR from isolated parasites generates bias as in vitro culture selects for strains growing faster in the selected medium, and genotyping by PCR only detects the dominant strains in potential mixtures of strains. More sensitive approaches based on direct genotyping by sequencing, and particularly deep sequencing, have shown that infections with multiple strains/DTUs can be frequent in some regions (see for example Villanueva-Lizama et al., J Infect Dis, 2019. 219(12): 1980-1988) and may have been overlooked in this study. These aspects should at least be discussed.

Answer: We agree with the reviewer regarding this limitation and we had already pointed out in the original submission that the blood culture method could select a specific T. cruzi subpopulation more adapted to the culture medium. We acknowledge that new sequencing methods could identify a higher percentage of mixed infections. However, the multilocus PCR used by us is also able to identify mixed infections and is recommended by experts’ consensus statement to be used in clinical samples [9]. In the revised manuscript, we created a limitations subheading in order to deepen this discussion, as suggested by the reviewer. Please, see page 19, line 485 to page 20, line 499 of the revised manuscript. 

Minor comments:

Question Follow-up time should be indicated for each patient in Table 4. 

Answer: The requested information was added. Please, see Table 4 of the revised manuscript. 

Question: Also, were any patient’s samples genotyped at more than 1 time point during their follow-up? If yes, were there any changes in parasite genotypes over time as seen in some studies? 

Answer: Patients’ samples were not genotyped at more than 1 time point. This limitation was added to the revised manuscript. Please, see page 19, line 489 to page 20, line 491 of the revised manuscript. 

Question: Has parasitemia been quantified in these patients? 

Answer: Parasitemia has not been quantified. 

Question:Were the patients treated? If yes, how did they respond to treatment? These aspects would enrich the study.

Answer: Few patients (eight) were treated with benznidazole during the study follow-up and the patients did not collect a new blood culture sample after treatment. Therefore, we could not study how specific treatment influenced the outcome of the studied patients.

The low number of treated patients in our study is due to the fact that most patients presented the chronic cardiac or cardiodigestive forms at the time of Chagas disease diagnosis and specific Chagas disease treatment would not be indicated in these cases. Also, the indication for specific Chagas disease treatment in patients with the indeterminate form was controversial at the time these patients were diagnosed and only recently there was a consensus towards specific Chagas disease treatment for chronic patients with the indeterminate form under 50 years old [3]. 

Most of the patients referred for treatment with benznidazole in the present study did so because they were enrolled in a research protocol (BENEFIT) that was underway during the study follow-up at our institution. BENEFIT recruited patients with the chronic cardiac form to test the clinical efficacy of benznidazol treatment. Therefore, most of the patients treated with benznidazol in our paper had the cardiac form. 

Please, see page 15, lines 356 to 358 for changes in the revised manuscript.

---

## [Editor Report · Decision Letter 1]

10 Nov 2020

PONE-D-20-26324R1

Association between Trypanosoma cruzi DTU TcII and chronic Chagas disease clinical presentation and outcome in an urban cohort in Brazil.

PLOS ONE

Dear Dr. Roberto Saraiva,

Thank you for submitting your manuscript to PLOS ONE. After careful consideration, we feel that it has merit for publication with a couple of minor edits. Therefore, we invite you to submit a revised version of the manuscript that addresses the points raised during the review process.

Editor Comments:

The authors well addressed the reviewers' comments. For the final acceptance of the article, I'm asking authors make a couple of minor edits on page 22, line 434, change "Limitations" by "Study limitations," and on page 18, line 349, replace "in the present article" with " in this study. "

We look forward to receiving your revised manuscript.

Kind regards,

Claudia Patricia Herrera, Ph.D.

Academic Editor

PLOS ONE

---

## [Author Response · Author response to Decision Letter 1]

10 Nov 2020

Editor Comments:

The authors well addressed the reviewers' comments. For the final acceptance of the article, I'm asking authors make a couple of minor edits on page 22, line 434, change "Limitations" by "Study limitations," and on page 18, line 349, replace "in the present article" with " in this study. "

- Answer: We made the requested changes. Please, see page 16 lines 294 to 295; and page 19, line 366 of the revised manuscript.

---

## [Editor Report · Decision Letter 2]

13 Nov 2020

Association between Trypanosoma cruzi DTU TcII and chronic Chagas disease clinical presentation and outcome in an urban cohort in Brazil.

PONE-D-20-26324R2

Dear Dr. Saraiva,

We’re pleased to inform you that your manuscript has been judged scientifically suitable for publication and will be formally accepted for publication once it meets all outstanding technical requirements.

Kind regards,

Claudia Patricia Herrera, Ph.D.

Academic Editor

PLOS ONE
---

## [Editor Report · Acceptance letter]

20 Nov 2020

PONE-D-20-26324R2 

Association between *Trypanosoma cruzi* DTU TcII and chronic Chagas disease clinical presentation and outcome in an urban cohort in Brazil. 

Dear Dr. Saraiva:

I'm pleased to inform you that your manuscript has been deemed suitable for publication in PLOS ONE. Congratulations! Your manuscript is now with our production department. 

Kind regards, 

on behalf of

Dr. Claudia Patricia Herrera 

Academic Editor

PLOS ONE